# Assessing the Association between Important Dietary Habits and Osteoporosis: A Genetic Correlation and Two-Sample Mendelian Randomization Study

**DOI:** 10.3390/nu14132656

**Published:** 2022-06-27

**Authors:** Jiawen Xu, Shuai Li, Yi Zeng, Haibo Si, Yuangang Wu, Shaoyun Zhang, Bin Shen

**Affiliations:** Orthopedic Research Institute, Department of Orthopedics, Sichuan University West China Hospital, Chengdu 610041, China; xujiawen_1997@163.com (J.X.); docleisure@163.com (S.L.); zengyigd@126.com (Y.Z.); sihaibo1987@163.com (H.S.); wuyuangang23@163.com (Y.W.); zhangsyhm2068@163.com (S.Z.)

**Keywords:** osteoporosis, dietary habits, linkage disequilibrium score regression, mendelian randomization

## Abstract

Objective: Osteoporosis (OP) is the most common bone disease. The genetic and metabolic factors play important roles in OP development. However, the genetic basis of OP is still elusive. The study aimed to explore the relationships between OP and dietary habits. Methods: This study used large-scale genome-wide association study (GWAS) summary statistics from the UK Biobank to explore potential associations between OP and 143 dietary habits. The GWAS summary data of OP included 9434 self-reported OP cases and 444,941 controls, and the GWAS summary data of the dietary habits included 455,146 participants of European ancestry. Linkage disequilibrium score regression (LDSC) was used to detect the genetic correlations between OP and each of the 143 dietary habits, followed by Mendelian randomization (MR) analysis to further assess the causal relationship between OP and candidate dietary habits identified by LDSC. Results: The LDSC analysis identified seven candidate dietary habits that showed genetic associations with OP including cereal type such as biscuit cereal (coefficient = −0.1693, *p* value = 0.0183), servings of raw vegetables per day (coefficient = 0.0837, *p* value = 0.0379), and spirits measured per month (coefficient = 0.115, *p* value = 0.0353). MR analysis found that OP and PC17 (butter) (odds ratio [OR] = 0.974, 95% confidence interval [CI] = (0.973, 0.976), *p* value = 0.000970), PC35 (decaffeinated coffee) (OR = 0.985, 95% CI = (0.983, 0.987), *p* value = 0.00126), PC36 (overall processed meat intake) (OR = 1.035, 95% CI = (1.033, 1.037), *p* value = 0.000976), PC39 (spirits measured per month) (OR = 1.014, 95% CI = (1.011, 1.015), *p* value = 0.00153), and servings of raw vegetables per day (OR = 0.978, 95% CI = (0.977, 0.979), *p* value = 0.000563) were clearly causal. Conclusions: Our findings provide new clues for understanding the genetic mechanisms of OP, which focus on the possible role of dietary habits in OP pathogenesis.

## 1. Introduction

Osteoporosis (OP) is a common disease and a major public health problem worldwide, with over nine million osteoporosis-related fractures occurring per year and is associated with high morbidity and mortality [1]. It is a strongly polygenetic disease, characterized by lumbar spine or hip bone mineral density (BMD) values that are at least 2.5 standard deviations below the population average in young healthy individuals [2]. In the United States, 10.3% or 10.2 million adults aged 50 and older have osteoporosis at the femoral neck or lumbar spine [3]. 

Genetic factors play a vital role in the pathogenesis of OP. There is strong evidence for a genetic predisposition to osteoporosis, with an estimated 60–80% of the risk explained by heritable factors. Recent studies have shown that approximately 40% to 80% of OP can be explained by genetic effects [4]. A new study has shown that after one week of a high-fat diet, bone metabolism is already altered in healthy men [5], which indicates that OP is influenced not only by genetic factors but also by environmental factors. However, to our knowledge, there is no study that has systematically explored the potential correlations between the dietary patterns and OP thus far. Genome-wide association studies (GWASs) of dietary habit profiling have provided unprecedented insights into how genetic variation influences dietary habits and complex diseases [6]. However, the genetic correlations and causal relationships between OP and dietary habits remain unclear. 

Linkage disequilibrium score regression (LDSC) is a widely used method to identify the genetic correlations among complex traits and to distinguish between inflated test statistics from confounding bias and polygenicity in GWAS [6]. Mendelian randomization (MR) analysis integrates genetic variants as instrumental variables (IVs) and evaluates the association between exposure and outcome using summary-level data from observational studies; MR has been used to identify reliable risk factors for various diseases [7,8]. LDSC combined with MR analysis has been widely used to explore the associations between complex diseases and their risk factors [7,8].

In the present study, we applied LDSC to detect genetic correlations between OP and each of the 143 dietary habits. MR analysis was then used to assess the causal relationship between OP and the candidate dietary habits identified by LDSC. Our study may elucidate the potential genetic mechanisms between dietary habits and OP.

## 2. Methods

### 2.1. GWAS Summary Data of OP

The GWAS summary data of OP were obtained from the UK Biobank resource [9]. The UK Biobank study is a large prospective cohort study of approximately 454,375 individuals aged between 37 and 76 years from all over the UK. The GWAS summary of OP included 9434 self-reported OP cases and 444,941 control participants. All participants provided a range of information on the demographics, health status, and lifestyle via questionnaires. Total body bone mineral density (TB-BMD) (g/cm^2^) was measured by dual-energy X-ray absorptiometry (DXA) using array beam mode QDR 2000, 2000 + bone densitometers, Hologic QDR 4500 workstations, or Hologic QDR 2000 workstations following the standard manufacturer’s protocols. Further details can be found in the UK Biobank fields: 20002.

### 2.2. GWAS Summary Data of Dietary Habits

Recent large-scale GWAS data of dietary habits were used here [10]. Briefly, phenotype derivation and genomic analysis were conducted on a homogenous population of 455,146 participants of European ancestry. LDstore v1.157 was used to calculate linkage disequilibrium (LD) and to identify SNPs in high LD (r^2^ ≥ 0.80) with any of the 77,229, 95% credible set SNPs. We used a strict Bonferroni correction threshold for all pair-wise tests between 143 dietary habits and 3219 highly correlated and even overlapping Neale Lab GWAS traits (*p* < 0.05/460,317 = 1.09 × 10^−7^). The linear mixed GWAS models were conducted on the 143 significantly heritable dietary habits (including both curated measures of single food intake (FI) and multivariate dietary patterns (DPs)) in up to 449,210 participants by using the Food Frequency Questionnaire (FFQ) data. FFQs offer useful information about the dietary intake and the relationship between it and the health and disease outcomes [11]. Due to their ease of administration, low burden on participants and staff, and lower cost compared to other assessment techniques, FFQs have been widely used in large population-based studies [12]. In total, 814 LD-independent loci (defined as >500 kb apart) were identified that surpassed genome-wide significance (*p* < 5.0 × 10^−8^). Detailed information on the study design, sample characteristics, quality control, and statistical analyses could be found in the published study [10].

### 2.3. Genetic Correlation Analysis

LDSC (v1.0.1, https://github.com/bulik/ldsc, accessed on 15 March 2022) software was applied to evaluate the genetic correlations between OP and each of the dietary habits. LDSC is a powerful approach for the genetic correlation analysis of complex diseases or traits [13]. LDSC can distinguish between true polygene and mixed biases (such as implicit association and demographic stratification) and is more effective than the genomic inflation factor (GIF, λ_GC_), especially in the case of a large sample size [13,14]. If the genetic association is statistically and quantitatively significant, we can be sure that the overall phenotypic association is not entirely attributed to environmental confounding factors [13]. In this study, we compared the relationships between the OP and 143 dietary habits. The significant association thresholds should be *p* < 0.000350 (0.05/143) after strict Bonferroni correction. *p* values between 0.000350 and 0.05 were considered to be suggestive of significance.

### 2.4. MR Analysis

MR analysis refers to the use of genetic variants in observational epidemiology to infer the variable risk factors for the disease and health-related outcomes [7]. In this study, MR analysis was used to evaluate the causal relationship between dietary habits (exposure) and OP (outcome). We carried out inverse variance weighted (IVW) methods. The significant dietary patterns identified by LDSC analysis were checked and included in the subsequent analysis. The SNPs were included as instrumental variables after filtering out SNPs whose distance was within 10,000 kb and r^2^ > 0.001. The number of SNPs included and the effect values (confidence intervals) and *p* values were reported. Moreover, to test the validity of our IVW results, heterogeneity and multiple validity tests were conducted by weighted median estimation and MR–Egger regression. In this study, we used TwoSampleMR packages (version 0.5.6) to perform the MR analysis in R (version 4.0.4). The statistical significance level was set at *p* < 0.05.

## 3. Results

### 3.1. Genetic Correlations between OP and Dietary Habits

The LDSC analysis identified seven candidate dietary habits showing suggestive association signals with OP such as biscuit cereal vs. any other (rg = −0.1693, *p* value = 0.0183), PC17 (butter) (rg = −0.136, *p* value = 0.0145), PC35 (decaffeinated coffee) (rg = −0.1367, *p* value = 0.0114), and PC36 (overall processed meat intake) (rg = −0.1443, *p* value = 0.0402). The overall results are summarized in Table 1. PC is a dietary pattern based on a specific diet. A specific description of PC can be found in a previous study [10] including biscuit cereal, PC17 (butter), PC35 (decaffeinated coffee), PC36 (overall processed meat intake), PC39 (spirits measured per month), servings of raw vegetables per day, and spirits measured per month. The all genetic correlations between 143 dietary habits and OP are summarized in Appendix A. 

### 3.2. Causal Relationships between OP and Dietary Habits

We identified causal relationships between the candidate dietary habits and OP such as PC17 (butter) (odds ratio [OR] = 0.974, 95% confidence interval [CI] = (0.973, 0.976), *p* value = 0.000970), PC35 (decaffeinated coffee (OR = 0.985, 95% CI = (0.983, 0.987), *p* value = 0.00126), PC36 (overall processed meat intake) (OR = 1.035, 95% CI = (1.033, 1.037) *p* value = 0.000976), PC39 (spirits measured per month) (OR = 1.014, 95% CI = (1.011, 1.015) *p* value = 0.00153), and servings of raw vegetables per day (OR = 0.978, 95% CI = (0.977, 0.979) *p* value = 0.000563) (Table 2, Appendix A).

## 4. Discussion

Our study aimed to evaluate the genetic correlations and causal relationships between the dietary habits and OP. First, by using the GWAS summary data of OP and 143 heritable dietary habits, we conducted a LDSC analysis to evaluate the genetic correlation between each of the dietary habits and OP. We identified seven candidate dietary habits showing suggestive association signals with OP. Second, we analyzed the causal relationships between the seven dietary habits and OP by MR analysis. We found that five dietary habits, namely, PC17 (butter), PC35 (decaffeinated coffee), PC36 (overall processed meat intake), PC39 (spirits measured per month), and servings of raw vegetables per day, showed a causal relationship with OP. Our study suggested that dietary habits play varied roles in the progression of OP. In fact, a systematic review provided an estimate of the association between different dietary patterns defined through the use of a posteriori methods and fracture or low BMD risk [15].

OP is a common disease and a major public health issue worldwide [1]. Recent research [16] has shown that chocolate is a rich source of antioxidant and anti-inflammatory flavonoids and dietary minerals with the potential to benefit bone health, and adolescents consuming chocolate had greater longitudinal bone growth. It is well-known that in the process of making chocolate, to promote its solidification, producers usually add butter [17]. A population-based trial [18] showed increased serum vitamin D levels and increased calcium absorption in men who consumed butter daily. Some studies have shown that the increased calcium absorption has a positive effect on BMD [19]. For example, a recent trial showed that after two years of high calcium intake levels, the BMD increased in female adolescents [20]. Our results are consistent with the above-mentioned existing studies, suggesting that the dietary behavior of consuming butter is a protective factor against OP.

Caffeine (1,3,7-trimethylxanthine) is a naturally occurring plant xanthine alkaloid present in many commonly consumed beverages worldwide including tea, coffee, and cocoa [21]. Moderate caffeine intake is generally considered to exert positive effects on human health such as the cardiovascular system and on the metabolism of carbohydrates and lipids [22]. A study showed the consumption of coffee was independently and significantly associated with OP, while the prevalence of OP was less frequent in Chinese men with moderate coffee intake [23]. However, inconsistent associations between coffee consumption and OP have been observed in other epidemiological studies [24]. Caffeine consumption appears to be associated with low BMD and fracture in several epidemiological studies [25]. A meta-analysis that included 7114 participants to investigate OP and thirteen studies with 391,956 participants investigating the fracture incidence suggested that a dose-dependent relationship may exist between coffee consumption and hip fracture incidence [26]. In addition, a relevant research showed that a daily intake of 330 mg of caffeine, which is equivalent to four cups (600 mL) of coffee, or more may be associated with a modestly increased risk of osteoporotic fractures, especially in women with a low intake of calcium [27]. The variations in clinical outcomes from these studies may be due to social and lifestyle factors as well as the different approaches to coffee production and consumption worldwide, and the wide range of other bioactive compounds in coffee [28]. In this study, we found that decaffeinated coffee was a protective factor for OP, which verified the view of studies that caffeine might be a risk factor for OP from another aspect.

Most processed meat products contain pork or beef but may also include other red meats, poultry, and meat byproducts such as animal offal or blood as well as hot processed sausage dogs, hams, sausages, beef jerky, canned meat, cold cuts of meat, and sauces. Recently, the World Health Organization (WHO) classified processed meat intake as “carcinogenic for humans” and red meat intake as “probably carcinogenic” [29]. A Western-type diet, characterized by a significant amount of highly processed and refined foods and high contents of sugars, salt, fat, and protein from red meat, has been recognized as an important factor contributing to the development of metabolic disorders and the obesity epidemic around the world [30]. According to a study [31], the Western diet was inversely related to BMD and directly related to osteoporosis risk. On the other hand, a dietary pattern that emphasized the intake of fruits, vegetables, whole grains, poultry and fish, nuts and legumes, and low-fat dairy products and deemphasized the intake of soft drinks, fried foods, meat and processed products, sweets and desserts, and refined grains showed a beneficial impact on bone health [31]. Dietary patterns represented by the Mediterranean diet, which is free of processed meat and contains a complex array of natural bioactive molecules with antioxidant, anti-inflammatory, and alkalizing properties, have been shown to reduce the incidence of OP [32].

Alcohol is widely consumed across the world in different cultural and social settings. The types of alcohol consumption can be classified as (a) light, only occasional consumption; (b) heavy chronic alcohol consumption; and (c) binge drinking, based on a new pattern of alcohol consumption [33]. Light to moderate alcohol consumption is generally reported to be beneficial, resulting in higher BMD and reduced age-related bone loss, whereas heavy alcohol consumption is generally associated with decreased BMD, impaired bone quality, and increased fracture risk [34]. Current research on the mechanisms of alcohol-induced bone loss including bone remodeling, bone immunity [35], and the gut microbiota [36] has focused on molecular mechanisms and cellular effects such as osteoblast apoptosis, oxidative stress, and regulation of the Wnt signaling pathway [37]. Our results also suggest that monthly spirits consumption is genetically associated with OP and may be a risk factor for OP.

Expert consensus and clinical practice guidelines state that adequate protein and vegetable intake is important to prevent OP [38]. Some polyphenol-rich foods including olive oil, fruits, and vegetables, tea and soy, seem to be beneficial for preventing OP and its progression [39]. A meta-analysis [40] evaluated the association of fruit and vegetable consumption and the risk of postmenopausal OP and suggested a significant association between the intake of vegetables and the risk of postmenopausal OP (OR, 0.62; 95% CI, 0.42–0.90). A study reported that a higher protein and dietary vitamin C intake was associated with a higher BMD, reduced risk of hip fracture, and slower rate of bone loss [38]. Nutrients, foods, and dietary patterns play an important role in maintaining bone health, and a balanced diet (including minerals, proteins, fruits and vegetables) is important for bone health and the prevention of fragility fractures [41].

This study detected the correlations between 143 dietary patterns and OP and found seven of them had potential correlations. In other words, this study did not observe correlations between the remaining 136 dietary patterns and OP temporarily. However, previous studies have found correlations between some of the remaining dietary patterns and OP. For example, some studies have suggested a positive link between milk intake and BMD and that the intake of milk can he protective against bone loss [42,43,44,45,46], which were inconsistent with our results. The reason for the inconsistent results may be due to the limited sample size and racial differences. The potential associations between remaining dietary patterns and OP will be explored in our subsequent studies by using different datasets.

This study had some limitations. First, the GWAS summary datasets were derived from published studies [9,10]. These two studies did not classify the population by specific factors such as age and sex. Therefore, our studies could not find any age- or sex-associated differences. Second, given the complexity of the pathogenesis of OP, we were unable to determine the specific role of these identified dietary habits in the pathogenesis of OP. The results of LDSC as well as MR only indicate possible genetic correlations and causal associations at the genetic level, and more mechanism-based experiments are needed to further confirm the biological rationality and to clarify the biological mechanism of the identified dietary habits expected to participate in the development of OP.

## 5. Conclusions

In summary, based on the GWAS summary data of OP and human dietary habits, we identified seven candidate dietary habits that were genetically correlated with OP through LDSC analysis. In addition, we explored the causal relationships between the seven dietary habits and OP by MR analysis and found that PC17 (butter), PC35 (decaffeinated coffee), PC36 (overall processed meat intake), PC39 (spirits measured per month), and servings of raw vegetables per day had a causal relationship with OP. Our study results provide novel clues for understanding the genetic mechanism of OP, focusing on the possible roles of abnormal dietary habits in the pathogenesis of OP.

## Figures and Tables

**Table 1 nutrients-14-02656-t001:** The genetic correlations between dietary habits and OP.

	Dietary Habits	Genetic Correlations	*p* Value
Osteoporosis	Biscuit cereal	−0.1693	0.0183
	PC17 (butter)	−0.1360	0.0145
	PC35 (decaffeinated coffee)	−0.1367	0.0114
	PC36 (overall processed meat intake)	−0.1443	0.0402
	PC39 (spirits measured per month)	0.1471	0.00990
	Servings of raw vegetables per day	0.0837	0.0379
	Spirits measured per month	0.1150	0.0353

Note: The LD score regression software (v1.0.1, https://github.com/bulik/ldsc, accessed on 15 March 2022) was used here to evaluate the genetic correlation between the dietary habits and osteoporosis (OP). OP: Osteoporosis.

**Table 2 nutrients-14-02656-t002:** The causal analysis results between the dietary habits and OP.

Exposure	Outcome	Number of SNP	Method	OR (95% CI)	*p* Value
PC17 (butter)	Osteoporosis	378	IVW	0.974 (0.973, 0.976)	0.000970
			WM		0.00126
			MR Egger		0.0130
PC35 (decaffeinated coffee)	Osteoporosis	285	IVW	0.985 (0.983, 0.987)	0.00126
			WM		0.00183
			MR Egger		0.00785
PC36 (overall processed meat intake)	Osteoporosis	1521	IVW	1.035 (1.033, 1.037)	0.000976
			WM		0.00100
			MR Egger		0.00332
PC39 (spirits measured per month)	Osteoporosis	157	IVW	1.014 (1.011, 1.015)	0.00153
			WM		0.00197
			MR Egger		0.0173
Servings of raw vegetables per day	Osteoporosis	1860	IVW	0.978 (0.977, 0.979)	0.000563
			WM		0.000712
			MR Egger		0.00341

Note: The MR analysis was performed through the TwoSampleMR packages (version 0.5.6) in R (version 4.0.4). The statistical significance level was set at *p* < 0.05. OP: Osteoporosis; MR analysis: Mendelian randomization analysis; IVW: Inverse variance weighted; WM: weighted median; OR: Odds ratio; CI: Confidence interval.

## Data Availability

The large-scale LDSC scan for potential genetic correlations between the OP and dietary habits was performed following the document of the LDSC tool (v1.0.1, https://github.com/bulik/ldsc, accessed on 15 March 2022). The MR analysis was performed through the TwoSampleMR packages (version 0.5.6).

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
