# Peer review of "Assessing the Association between Important Dietary Habits and Osteoporosis: A Genetic Correlation and Two-Sample Mendelian Randomization Study"

_nutrients, 2022, doi:10.3390/nu14132656_

Round 1

Reviewer 1 Report

Xu et al., have used the data in the UK Biobank to associate dietary data with low bone mass (BMD) and genetics.  The authors use linkage disequilibrium to associate alleles to bone mass, and the Mendelian randomization to identify dietary risk factors  associated with low BMD.  The manuscript overall, is well written and the results are unexpected and compelling. There are a few suggestions to improve the data presentation and clarify intent (see below).  

Some minor comments: 

1) Vegetables consumption are usually in servings (1 serving= 1 cup), not tablespoons. Please clarify units.

2) In results the authors use the term "cereal type cereal biscuits";  this seems redundant, perhaps the authors mean cereal biscuits (like Weetabix) ?  Please clarify (I appreciate this is likely cultural issue, because this term is very British). 

3) Methods, the authors provide the version or date of the LDSC software, used in the analysis.

3) Discussion: the MR method identifies risk factors, but authors find association between foods (or food groups) and often explain results are protective of bone mass.  However, this is confusing to me. They can either describe foods with increased or decreased risk of low BMD?  Please clarify. 

Author Response

  1. Vegetables consumption are usually in servings (1 serving= 1 cup), not tablespoons. Please clarify units.

Response: Thank you for the helpful comments.

We are very sorry for the confused places. Per your guidance, we have corrected “tablespoons of raw vegetables per day” into “servings of raw vegetables per day”. Please see in the revised manuscript.

Thank you!

  1. In results the authors use the term "cereal type cereal biscuits";  this seems redundant, perhaps the authors mean cereal biscuits (like Weetabix) ?  Please clarify (I appreciate this is likely cultural issue, because this term is very British). 

Response: Thank you for the helpful comments.

We are very sorry for the confused places. In this study, candidate dietary habits “cereal type: cereal biscuits” means the type of cereal is biscuits. The phrase “cereal type: cereal biscuits” was quoted to previous study [1]. Per your guidance, we have corrected “cereal type: cereal biscuits” into “biscuits cereal”. Please see in the revised manuscript.

Thank you!

  1. Methods, the authors provide the version or date of the LDSC software, used in the analysis.

Response: Thank you for the helpful comments.

We are very sorry for the confused places. In this study, the version of LDSC software used was v1.0.1 (https://github.com/bulik/ldsc), which were accessed on 15 March 2022. Per your guidance, we added the version of the LDSC software and the date accessed website in the method section. Please see page 3, line 103 in the revised manuscript.

 Thank you!

  1. Discussion: the MR method identifies risk factors, but authors find association between foods (or food groups) and often explain results are protective of bone mass.  However, this is confusing to me. They can either describe foods with increased or decreased risk of low BMD?  Please clarify. 

Response: Thank you for the helpful comments.

We are very sorry for the confused places. We have removed the inappropriate discussion of the foods are protective of bone mass. Per your guidance, we added some evidence which support the relationships between different dietary patterns and its increased or decreased risk of BMD in the discussion section. This can further prove the correlation between dietary patterns and OP risks. Please see page 2, 5-6, line 44-48, 174-177, 186-188, 207-208, 234-235 in the revised manuscript.

Reviewer 2 Report

Xu et. al. report about the association between dietary habits and osteoporosis using a huge database including more than 9,500 osteoporosis patients and more than 445,000 healthy controls. They focused on 143 dietary habits and found 7 candidates showing genetic associations with osteoporosis. The topic is relevant and the methods appropriate. However, the manuscript has major limitations. 

Major limitations:

The authors talk about 143 nutritional habits, but only report 7 of them. What are the remaining 136 factors? I suggest to include a table with all the nutritional habits and also discuss well known and non-significant factors, e.g. protein, phosphate, diaries? 

What kind of questionnaire has been used? Is it validated? Have you considered the frequency of food consumption?

One of the 7 candidates is decaffeinated coffee with an odd ratio of 0.985. The discussion focuses only on the influence of caffeine to the bone metabolism. 

How did you define osteoporosis? Did they suffer from fractures? Was there an association between BMD and nutritional habits?

Did you find any age- or sex-associated differences?

What kind of other risk factors where available in your database, e.g. smoking, medication (glucocorticoids, PPI, …), exercise, vitamin D blood values, etc.?

Regarding the study design and sample characteristics: In a study like this, it´s not appropriate to state “are described elsewhere”. Please give more information about the design and study group as well as the methods used.

Minor limitations:

Regarding to the 7 candidate dietary habits spirits measured per month are mentioned two times. Please clarify.

Please change statistical significance from e.g. 1.83E-0.2 to p=0.0183 as it is the scientific way.

In the Introduction you mentioned, that “there are no systematic studies to discover the potential correlations between dietary patterns and OP”. In the discussion section, various studies to dietary influences on osteoporosis are cited. Please change.

Author Response

Major limitations:

1.The authors talk about 143 nutritional habits, but only report 7 of them. What are the remaining 136 factors? I suggest to include a table with all the nutritional habits and also discuss well known and non-significant factors, e.g. protein, phosphate, diaries?

Response: Thank you for the helpful comments.

We are very sorry for the confused places. This study detected the correlations between 143 dietary patterns and OP and found 7 of them had potential correlations. LDSC analysis did not observed correlations between the remaining 136 dietary patterns and OP temporarily. Per your guidance, we added a supplemental table to show all the results and added some discussion of well-known and non-significant factors. Please see page 6, line 239-247 and supplemental table 1 in the revised manuscript.

This study detected the correlations between 143 dietary patterns and OP and found 7 of them had potential correlations. In other words, this study did not observed correlations between the remaining 136 dietary patterns and OP temporarily. But previous studies have found correlations between some of the remaining dietary patterns and OP. For example, some studies have suggested a positive link between milk intake and BMD, and intaking milk can protective against bone loss [1-5], which were inconsistent with our results. The reason for the inconsistent results may be due to the limited sample size and racial differences. The potential associations between remaining dietary patterns and OP will be explored in our subsequent studies by using different data sets.

In addition, these 143 dietary patterns were classified by the type of food instead of content of nutrients. In other words, the current study could not discuss the correlations between OP and nutrients such as protein, phosphate, and diaries. The potential associations between these factors and OP will be explored in our subsequent studies.

Thank you!

2.What kind of questionnaire has been used? Is it validated? Have you considered the frequency of food consumption?

Response: Thank you for the helpful comments.

We are sorry for the confused places. The questionnaire used in previous GWAS of dietary habits is Food Frequency Questionnaire (FFQ) [6]. Its effectiveness has been verified in many studies [7]. The usual food intakes derived from the FFQ were calculated by multiplying the frequency of consumption with a standard portion size for each food item. Therefore, the frequency of food consumption is well reflected in this questionnaire [8]. Per your guidance, we have added the details of the questionnaire used in Methods section. Please see page 2-3, line 94-98 in the revised manuscript.

Thank you!

3.One of the 7 candidates is decaffeinated coffee with an odd ratio of 0.985. The discussion focuses only on the influence of caffeine to the bone metabolism.

Response: Thank you for the helpful comments.

We are very sorry for the confused places. There are inconsistent associations between coffee consumption and OP have been observed in previous epidemiological studies. The possible mechanism may be a dose-dependent relationship exists between coffee consumption and OP. Per your guidance, we added a detailed discussion on the influence of caffeine to OP. Please see the following description and page 5, line 179-198 in the revised manuscript.

Caffeine (1,3,7-trimethylxanthine) is a naturally occurring plant xanthine alkaloid present in many commonly consumed beverages worldwide, including tea, coffee, and cocoa [9]. Moderate caffeine intake is generally considered to exert positive effects on human health, such as the cardiovascular system, and on the metabolism of carbohydrates and lipids [10]. A study showed the consumption of coffee was independently and significantly associated with OP, while the prevalence of OP was less frequent in Chinese men with moderate coffee intake [11]. However, inconsistent associations between coffee consumption and OP have been observed in other epidemiological studies [12]. Caffeine consumption appears to be associated with low BMD and fracture in several epidemiological studies. A meta-analysis included 7,114 participants for investigating OP and thirteen studies with 391,956 participants for investigating fracture incidence suggested that a dose-dependent relationship may exist between coffee consumption and hip fracture incidence [13]. In addition, a relevant research showed that a daily intake of 330 mg of caffeine, which equivalents to 4 cups (600 ml) of coffee, or more may be associated with a modestly increased risk of osteoporotic fractures, especially in women with a low intake of calcium [14].The variations in clinical outcomes from these studies may be due to social and lifestyle factors, as well as the different approaches to coffee production and consumption worldwide, and the wide range of other bioactive compounds in coffee [15]. In this study, we found that decaffeinated coffee was a protective factor for OP, which verified the view studies that caffeine might be a risk factor for OP from another aspect.

Thank you!

4.How did you define osteoporosis? Did they suffer from fractures? Was there an association between BMD and nutritional habits?

Response: Thank you for the helpful comments.

We are very sorry for the confused places. Osteoporosis (OP) is a systemic skeletal disease characterized by low bone mass and microarchitectural deterioration in bone tissue, leading to enhanced bone fragility and increased fracture risk [16]. In the UK Biobank, OP was defined by the ICD 10 code which were collected from different source of report on M81 (osteoporosis without pathological fracture). So, all those OP participates included are not suffering from pathological fracture.

Bone mineral density (BMD) is used conventionally as a proxy for overall bone strength and is expressed as grams of mineral per square centimeter or grams per cubic centimeter [17]. Patients with OP are usually associated with a decrease in BMD. When the hip BMD T score is -2.5 or less, it is considered osteoporosis [18]. Our study has identified 7 dietary habits related to the OP, so there is a potential association between dietary habits and BMD. This potential association will be confirmed in our subsequent studies.

Thank you!

  1. Did you find any age- or sex-associated differences?

Response: Thank you for the helpful comments.

We are very sorry for the confused places. The GWAS summary data sets were derived from previous published studies [19,20]. These two studies did not classify the population by specific factors such as age and sex. So, our studies could not find any age- or sex-associated differences. Per your guidance, we added this as a limitation in the discussion section. Please see page 6, line 248-251 in the revised manuscript.

Thank you!

6.What kind of other risk factors where available in your database, e.g. smoking, medication (glucocorticoids, PPI, …), exercise, vitamin D blood values, etc.?

Response: Thank you for the helpful comments.

We are very sorry for the confused places. The GWAS summary data of exposure was derived from published studies [20]. This GWAS summary data contained 143 dietary patterns. Per your guidance, we added a supplemental table to show all well-known and non-significant factors. Please see supplemental table 1 in the revised manuscript. Unfortunately, this GWAS summary data did not contain other factors such as smoking, medication, exercise, and vitamin D blood values. The potential associations between these factors and OP will be explored in our future studies.

Thank you!

7.Regarding the study design and sample characteristics: In a study like this, it´s not appropriate to state “are described elsewhere”. Please give more information about the design and study group as well as the methods used.

Response: Thank you for the helpful comments.

We are very sorry for the confused places. We added the more information about the design and study group in the method section. Please see page 2-3, line 89-101 in the revised manuscript.

Thank you!

Minor limitations:

1.Regarding to the 7 candidate dietary habits spirits measured per month are mentioned two times. Please clarify.

Response: Thank you for the helpful comments.

We are very sorry for the confused places. In this study, LDSC analysis identified 7 candidate dietary habits included principal component (PC) 39 (spirits measured per month) and spirits measured per month. Although single dietary patterns and multivariate dietary patterns (PC-DPs) were both from the published study[20], these two candidate dietary habits are not same. “Spirits measured per month” is a single dietary pattern. In this study, single dietary patterns were derived from FFQ. However, PC-DPs were generated by analyzing the principal component of single dietary patterns. In other words, PC-DP is a collection of single dietary patterns. PC39 is composed by 16 single dietary patterns such as spirits measured per month and servings of raw vegetables per day. Spirits measured per month contributed the most component of PC39 (approximately 20%). To summary, this study proved a potential correlation between spirits measured per month and OP at the single and collective levels.

Thank you!

2.Please change statistical significance from e.g. 1.83E-0.2 to p=0.0183 as it is the scientific way.

Response: Thank you for the helpful comments.

We are very sorry for the confused place. Per your guidance, we have changed all the statistical significances format in a scientific way. Please see in the revised manuscript.

Thank you!

3.In the Introduction you mentioned, that “there are no systematic studies to discover the potential correlations between dietary patterns and OP”. In the discussion section, various studies to dietary influences on osteoporosis are cited. Please change.

Response: Thank you for the helpful comments.

We are very sorry for the confused places. Studies which quoted in the discussion section did have discussed the potential correlations between dietary patterns and OP. However, most of them only discussed the single dietary pattern and OP. This study is the first time to discover the potential correlations between dietary patterns and OP systematically. Per your guidance, we rephrased this description. Please see page 2, line 55-56 in the revised manuscript.

Thank you!

Reference

  1. Sahni S, Mangano KM, Kiel DP, Tucker KL, Hannan MT. Dairy Intake Is Protective against Bone Loss in Older Vitamin D Supplement Users: The Framingham Study. J Nutr. 2017;147(4):645-652.
  2. Thorpe MP, Jacobson EH, Layman DK, He X, Kris-Etherton PM, Evans EM. A diet high in protein, dairy, and calcium attenuates bone loss over twelve months of weight loss and maintenance relative to a conventional high-carbohydrate diet in adults. J Nutr.2008;138(6):1096-1100.
  3. Moschonis G, Manios Y. Skeletal site-dependent response of bone mineral density and quantitative ultrasound parameters following a 12-month dietary intervention using dairy products fortified with calcium and vitamin D: the Postmenopausal Health Study. Br J Nutr. 2006;96(6):1140-1148.
  4. Polzonetti V, Pucciarelli S, Vincenzetti S, Polidori P. Dietary Intake of Vitamin D from Dairy Products Reduces the Risk of Osteoporosis. Nutrients. 2020;12(6):1743.
  5. Sahni S, Tucker KL, Kiel DP, Quach L, Casey VA, Hannan MT. Milk and yogurt consumption are linked with higher bone mineral density but not with hip fracture: the Framingham Offspring Study.Arch Osteoporos. 2013;8(0):119.
  6. De Keyzer W.; Dekkers A.; Van Vlaslaer V.; Ottevaere C.; Van Oyen H.; De Henauw S, Huybrechts I. Relative validity of a short qualitative food frequency questionnaire for use in food consumption surveys. Eur J Public Health.2013, 23,737-742.
  7. Vijay A, Mohan L, Taylor MA, et al. The Evaluation and Use of a Food Frequency Questionnaire Among the Population in Trivandrum, South Kerala, India. Nutrients. 2020;12(2):383
  8. Clarys P.; Deliens T.; Huybrechts I.; Deriemaeker P.; Vanaelst B.; De Keyzer W.; Hebbelinck M, Mullie P. Comparison of nutritional quality of the vegan, vegetarian, semi-vegetarian, pesco-vegetarian and omnivorous diet. 2014, 6,1318-1332.
  9. Xu H.; Liu T.; Hu L.; Li J.; Gan C.; Xu J.; Chen F.; Xiang Z.; Wang X, Sheng J. Effect of caffeine on ovariectomy-induced osteoporosis in rats. Biomed Pharmacother.2019, 112,108650.
  10. Cano-Marquina A.; Tarín JJ, Cano A. The impact of coffee on health. Maturitas.2013, 75,7-21.
  11. Yu Q.; Liu ZH.; Lei T, Tang Z. Subjective evaluation of the frequency of coffee intake and relationship to osteoporosis in Chinese men. J Health Popul Nutr.2016, 35,24.
  12. Chau YP.; Au PCM.; Li GHY.; Sing CW.; Cheng VKF.; Tan KCB.; Kung AWC, Cheung CL. Serum Metabolome of Coffee Consumption and its Association With Bone Mineral Density: The Hong Kong Osteoporosis Study. J Clin Endocrinol Metab.2020, 105.
  13. Zeng X.; Su Y.; Tan A.; Zou L.; Zha W.; Yi S.; Lv Y, Kwok T. The association of coffee consumption with the risk of osteoporosis and fractures: a systematic review and meta-analysis. Osteoporos Int.2022.
  14. Hallstrom H.; Wolk A.; Glynn A, Michaelsson K. Coffee, tea and caffeine consumption in relation to osteoporotic fracture risk in a cohort of Swedish women. Osteoporos Int.2006, 17,1055-1064.
  15. Berman NK.; Honig S.; Cronstein BN, Pillinger MH. The effects of caffeine on bone mineral density and fracture risk. Osteoporos Int.2022, 33,1235-1241.
  16. Ensrud KE, Crandall CJ. Osteoporosis. Ann Intern Med.2017, 167,Itc17-itc32.
  17. Lane NE. Epidemiology, etiology, and diagnosis of osteoporosis. Am J Obstet Gynecol.2006, 194,S3-11.
  18. 1 Cotts KG, Cifu AS. Treatment of Osteoporosis. JAMA. 2018;319(10):1040-1041.
  19. Ollier W.; Sprosen T, Peakman T. UK Biobank: from concept to reality. Pharmacogenomics.2005, 6,639-646.
  20. Cole JB.; Florez JC, Hirschhorn JN. Comprehensive genomic analysis of dietary habits in UK Biobank identifies hundreds of genetic associations. Nat Commun.2020, 11,1467.
